# Collocation Mismatch Uncertainties in Satellite Aerosol Retrieval Validation

Timo H. Virtanen[1], Pekka Kolmonen[1], Larisa Sogacheva[1], Edith Rodríguez[1], Giulia Saponaro[1], and Gerrit de Leeuw[1]

[1]Finnish Meteorological Institute, Helsinki, Finland

*Correspondence to:* T. H. Virtanen (timo.h.virtanen@fmi.fi)

**Abstract.** Satellite based aerosol products are routinely validated against ground based reference data, usually obtained from sunphotometer networks such as AERONET (AEROsol Robotic Network). In a typical validation exercise a spatial sample of the instantaneous satellite data is compared against a temporal sample of the point-like ground based data. The observations do not correspond to exactly the same column of the atmosphere at the same time, and the representativiness of the reference data depends on the spatiotemporal variability of the aerosol properties in the samples. The associated uncertainty is known as the collocation mismatch uncertainty (CMU). The validation results depend on the sampling parameters. While small samples involve less variability, they are more sensitive to the inevitable noise in the measurement data. In this paper we study systematically the effect of the sampling parameters in the validation of AATSR (Advanced Along Track Scanning Radiometer) aerosol optical depth (AOD) product against AERONET data and the associated collocation mismatch uncertainty. To this end, we study the spatial AOD variability in the satellite data, compare it against the corresponding values obtained from densely located AERONET sites, and assess the possible reasons for observed differences.

We find that the spatial AOD variability in the satellite data is approximately two times larger than in the ground based data, and the spatial variability correlates only weakly with that of AERONET for short distances. We interpret that only half of the variability in the satellite data is due to the natural variability in the AOD, and the rest is noise due to retrieval errors. However, for larger distances ($\sim 0.5°$) the correlation is improved as the noise is averaged out, and the day to day changes in regional AOD variability are well captured. Furthermore, we assess the usefulness of the spatial variability of the satellite AOD data as an estimate of CMU by comparing the retrieval errors to the total uncertainty estimates including the CMU in the validation. We find that accounting for CMU increases the fraction of consistent observations.

## 1 Introduction

Satellite-based instruments are widely used to retrieve information on aerosols on a global scale. Aerosol retrieval algorithms for satellite based data involve several assumptions, and the retrieval results need to be carefully validated against ground based data. The validation of products with a typical resolution of several kilometers against point-like ground based measurements involves uncertainties. A key question is how well does the point-like ground-based measurement represent a larger area around

the measurement site. To assess this, we study the spatial variability of aerosol optical depth (AOD) in an area covering several satellite data points around the measurement site.

Our aim is to obtain information on the spatial variability of AOD using the satellite data only, so that it could be included in the satellite product as an estimate of the collocation mismatch uncertainty (CMU) without the need of auxiliary data sources. However, the satellite-based aerosol data can be noisy due to retrieval errors caused e.g. by residual clouds, varying surface reflectance and in case of dual view retrieval techniques by collocation errors between the two viewing directions. The AOD variability obtained from satellite data may contain a significant contribution from these errors and needs to be evaluated against ground-based data, such as the sunphotometer data obtained from AERONET (AEROsol Robotic Network). Usually the ground based data is not available on a spatial scale relevant to the AOD validation ($\lesssim 1°$), since the AERONET sites are located far from each other. Fortunately, there are campaigns that can provide ground based AOD data with sufficient spatial resolution, such as the AERONET DRAGON (Distributed Regional Aerosol Gridded Observational Network) campaign in the Baltimore region in summer 2011, which was part of the National Aeronautics and Space Administration's (NASA) DISCOVER-AQ (Deriving Information on Surface conditions from Column and Vertically Resolved Observations Relevant to Air Quality) field campaign. For a review of the DRAGON campaigns and studies based on them we refer to Holben et al. (2017) and references therein.

The spatial variability of AOD and the related issues in the validation of the satellite data against AERONET have been frequently studied since the first satellite AOD products became available. In particular, the sampling used in the validation, i.e. the spatial averaging of satellite data and temporal averaging of the AERONET data has been investigated. Ichoku et al. (2002) validated MODIS AOD data against AERONET and also compared the spatial variation of MODIS AOD to the temporal variation of AERONET AOD. They found that there is a correlation (R∼0.4) between the spatial standard deviations of the surface reflectance at 2.1 $\mu$m and AOD at 470nm and 660nm. This suggests that the variation in the satellite AOD is partly caused by varying surface reflectance (i.e. failure of the satellite retrieval algorithm to capture the true surface reflectance). In this work we study systematically the effect of the sampling parameters used in satellite AOD validation and the related standard deviations. Our focus is on the AATSR (Advanced Along Track Scanning Radiometer) data and the ADV (AATSR Dual View) algorithm Kolmonen et al. (2016), but we also apply the methods to MODIS data to reveal possible instrument specific effects.

The validation of MODIS AOD products is described by Levy et al. (2013) (dark target), Sayer et al. (2013, 2014) (deep blue), and Remer et al. (2013) (3 km product). Munchak et al. (2013) studied in detail the aerosol variability and the effect of MODIS retrieval resolution on the validation against DISCOVER-AQ field campaign data. When comparing the performance of the MODIS 3 km and 10 km AOD products against the AERONET DRAGON campaign they found that the 3 km product has better coverage and resolves the aerosol gradients better, but is noisier, especially in urban areas. From the High Spectral Resolution Lidar (HSRL) airborne lidar data collocated with satellite overpass Munchak et al. (2013) found that AOD can vary by more than 0.2 within a single 10 km pixel of the MODIS aerosol product, indicating that there are large uncertainties involved in validating large satellite footprints with the point-like AERONET measurements. The validation of MISR aerosol retrieval is discussed by Kahn et al. (2005, 2007, 2010). The comparison of a new 4.4 km MISR product against DRAGON

campaign data is discussed by Garay et al. (2017). They found that the new 4.4 km product performs better in comparison with the DRAGON data, which they attribute to the higher resolution algorithm being better able to capture the true spatial variability of aerosols.

Li et al. (2016) have studied the AERONET locations using multisensor satellite data and an ensemble Kalman filter approach. They analyzed the spatial representativeness of individual AERONET sites, and found that this depends on the season and the dominant aerosol type. Lee and Son (2016) and Sano et al. (2016) studied the variability of AERONET aerosol optical properties during the DRAGON-Asia campaign in 2012. Sano et al. (2016) concluded that due to the high variability in AOD, the ground based measurements should be more frequent, and the satellite retrievals should have a finer resolution for a proper comparison. Xiao et al. (2016) compared AOD retrievals from several satellites to data from the DRAGON-Asia campaign and handheld sunphotometers, and concluded that the satellite products are better at tracking the day-to-day variability than tracking the spatial variability.

The AERONET AOD data are commonly used as a reference data for satellite AOD products, and the associated uncertainty of 0.01-0.02 (Eck et al., 1999) is usually small compared to the corresponding uncertainties in the satellite retrievals. However, when validating satellite products against AERONET one should bear in mind that the AERONET data is not errorless, and even small uncertainties in the reference data may cause biases and affect the conclusions, especially when using regression analysis, as recently discussed by Pitkänen et al. (2016). In this paper we show linear regression lines on some plots, but avoid making far reaching conclusions based on these.

The ADV retrieval algorithm provides an AOD uncertainty estimate for each pixel, based on the propagation of the reflectance measurement uncertainty through the retrieval. Here we study the effect of the additional collocation mismatch uncertainty in the validation. It is difficult to assess the validity of uncertainty estimates. Two approaches are presented: the AOD correlation should be better for cases with lower CMU, and on average the AOD error should be less than the corresponding uncertainty. For the first case, we study the dependence of the AOD correlation coefficient on an AOD spatial standard deviation threshold. In the second case, we study the relationship between error and different uncertainty estimates, i.e. with or without the contribution of the CMU estimate.

The rest of the paper is structured as follows. In section 2 we briefly introduce the instruments used. In section 3 we discuss the relevant features of the ADV retrieval algorithm, and describe the methods used in the comparison. In section 4 we present and discuss the results of the satellite-AERONET comparison. Section 5 concludes the paper. A supplement with additional figures and tables is provided with this paper.

## 2   Instruments

### 2.1   AATSR

The European Space Agency's (ESA) Advanced Along Track Scanning Radiometer (AATSR) aboard the ENVISAT satellite measured the top of atmosphere (TOA) radiance at seven wavelengths ranging from the visible to the thermal infrared. The nominal AATSR resolution is 1 km (L1) and the swath width is ~500 km, which provided a revisit time of 3-4 days at mid-

latitudes for the ten-year mission (2002-2012). AATSR was a dual view instrument, scanning each pixel from a 55° forward and a near-nadir view. The ADV aerosol retrieval algorithm, employing the dual-view capability of AATSR, is described in section 3.1.

## 2.2 MODIS

While our main focus is on AATSR data, we also use MODerate resolution Imaging Spectroradiometer (MODIS) data to study the spatial aerosol variability near AERONET DRAGON sites. We use collection 6 AOD data and both the 10 km and 3 km aerosol products (see e.g. Remer et al. (2013)), and concentrate on the Terra satellite for a closer temporal match with AATSR. A more thorough comparison of MODIS and DISCOVER-AQ data has been done by Munchak et al. (2013), and this effort will not be repeated here. We focus on calculating the spatial standard deviation of AOD from MODIS Terra, and compare that to the results from AERONET and AATSR.

## 2.3 AERONET

AERONET (Aerosol Robotic Network) is a network of sun photometer instruments deployed at several hundred locations over the world for monitoring aerosols (Holben et al., 1998). The AERONET sun photometers measure solar irradiance at several wavelengths from UV to NIR, and provide AOD with an uncertainty of 0.01-0.02 (Eck et al., 1999). In this exercise we use the quality-assured, cloud-screened Level 2.0 AERONET AOD data for the wavelengths 440, 675, 870, and 1020 nm. Since these wavelengths do not match with those of the ADV aerosol product, Ångström exponent is used to derive AERONET AOD values at 555 and 659 nm wavelengths.

AERONET deployed more than 40 CIMEL Sun-sky radiometers in the Baltimore-Washington DC region in the summer 2011 for the DRAGON campaign, as part of DISCOVER-AQ campaign (Holben et al., 2017). While several other DRAGON measurement campaigns have been arranged since 2011, we are only able to use data from the 2011 campaign and part of the 2012 campaign (DRAGON Asia), since the connection to ENVISAT was lost in April 2012. In this paper we concentrate only on the 2011 campaign, and limit the AATSR data to the area limited to longitudes between 77.2° W and 75.8° W and to latitudes between 38.7° N and 39.8° N.

The 2011 DRAGON campaign provides a grid of AERONET sites with a roughly 10 km spacing, producing detailed information on aerosol spatial variability on a scale typical of satellite retrievals (Fig. 1a). We use the DRAGON observations to study the natural AOD variability, and to evaluate the collocation mismatch uncertainty estimate obtained from AATSR data. The region is interesting as it provides different aerosol loads on a surface varying from urban to agricultural areas as well as water. Figure 1 (b) illustrates the AOD variability in the area and the associated correlation length. For each day in June-August 2011 we calculated the daily correlation coefficient between each pair of sites for temporally collocated AOD observations. With the 37 sites in our study area and with 76 days of available data we obtained 36352 correlation coefficient values, when the cases with less than 5 simultaneous observations per day were excluded. These values are presented in Fig. 1 (b) as a function of the distance between the sites. As expected, the average correlation is high for short distances, but drops below

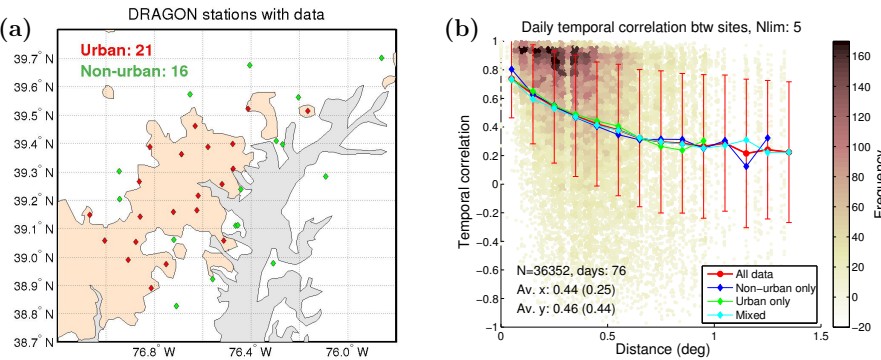

**Figure 1. (a)** Map of the DRAGON 2011 area, with 21 urban AERONET sites (red symbols) and 16 non-urban (green symbols) sites with data. The gray area is water (Chesapeake bay), and the light brown color shows urban areas. **(b)** Temporal correlation between different AERONET sites in the DRAGON area as function of the distance between the sites. The lines show average correlation for distance bins for different area types, and the error bars show the corresponding standard deviation. Differences between urban, non-urban and mixed (other site urban, other non-urban) are small.

0.4 for distances larger than $0.5°$. These results give a reference scale for the comparison between the AERONET and satellite based AOD values.

Munchak et al. (2013) report that the MODIS 3 km AOD product performs less well for urban areas. We used the urban area classification from Schneider et al. (2003) (Fig. 1 (a)) to study this, but did not find such trend in the AATSR data. Figure 1 (b) indicates that there are no significant differences between the AERONET sites in urban or rural areas either. However, an issue with the AATSR cloud screening for the highly reflecting urban areas was discovered. On some clear days one of the ADV cloud tests interpreted the bright urban surfaces as clouds. The cloud test was modified to allow more retrievals over the urban areas for this study.

## 3 Methods

### 3.1 ADV algorithm

The AATSR Dual View (ADV) algorithm is originally based on the work by Veefkind and de Leeuw (1998), and the current version is described by Kolmonen et al. (2016). The algorithm uses the AATSR stereo view to remove the surface reflectance contribution from the TOA reflectance and retrieves the best fit aerosol model and AOD value using inversion techniques. The ADV algorithm is used over land surfaces and the retrieval product provides AOD values at three wavelengths, 555 nm, 659 nm, and 1.6 $\mu$m.

The AATSR L1 data at 1 km resolution is first cloud screened, and resampled to a $0.1° \times 0.1°$ grid, which is used in the retrieval. Here we use the ADV v3.10 data, except that one of the cloud test has been slightly modified. It was discovered that the urban areas in Washington D.C. and Baltimore, which are brighter than the surroundings, were sometimes misidentified

as clouds for otherwise cloud free scenes. A lower threshold of 0.15 for cloud reflectance was forced to the brightness histogram cloud test employing the 659 nm channel to remove the misidentification issue. The modified cloud screening was then inspected visually for each orbit, and no signs of additional cloud contamination were observed.

After the AOD retrieval, a further cloud post-processing is applied to remove residual clouds and cloud edges. The post-processing is based on thresholds on the local AOD variability (standard deviation of AOD) and the number of neighboring cloud free pixels in a $3 \times 3$ pixels area (Sogacheva et al., 2017). ADV algorithm provides a per pixel AOD uncertainty estimate based on the propagation of the assumed 5% uncertainty in the measured reflectance through the retrieval (Kolmonen et al., 2016). This uncertainty estimate does not include sampling and smoothing uncertainties, uncertainties related to the selection of the best-fit aerosol model (Kauppi et al., 2017), or uncertainties related to the cloud screening. In this work we study the additional collocation mismatch uncertainty related to the validation against AERONET.

## 3.2 Comparison method

The AERONET quality-assured Level 2.0 AOD data is commonly used for validation of satellite-based aerosol products (e.g. Kahn et al. (2010); Levy et al. (2013); de Leeuw et al. (2015)). The simplest approach in validating satellite aerosol products against AERONET is to compare the single satellite pixel which encloses the AERONET site to the single AERONET measurement closest in time to the satellite overpass, but this is not necessarily the ideal method (Ichoku et al., 2002). Firstly, the satellite measurement always represents a spatial average over the pixel area with varying atmospheric and surface conditions, and the point-like AERONET measurement may not be representative of these conditions. Secondly, there is usually a time gap between the observations, and the observation axes of the measurements differ. Hence it is a common practice to compare spatial statistics of the satellite data to temporal statistics of the AERONET data.

Ichoku et al. (2002) tested the use of various sampling sizes (from 30 to 90 km squared) for the AOD validation, and found that the dependence of the mean AOD on the sampling window size is small and does not have a specific trend. They end up recommending a 50 km sampling area ($5 \times 5$ MODIS pixels), corresponding roughly to 1 h of AERONET data for an average aerosol travel velocity of 50 km/h. In the MODIS C6 validation a spatial radius of $\pm 25$km for satellite data ($\sim 25$ MODIS pixels) and a temporal window of $\pm 30$ min for AERONET is used (Levy et al., 2013). Munchak et al. (2013) studied the validation of both MODIS 3 km and 10 km products against DRAGON 2011 data, with a $5 \times 5$ pixel sampling area ($15 \times 15$ km$^2$ for the 3 km product and $50 \times 50$ km$^2$ for the 10 km product). They also tested single pixel validation (only use the satellite pixel containing the AERONET site) and found that the spatial averaging technique better characterizes the performance of the retrieval algorithm, and the single pixel method has larger representation uncertainty since it is more sensitive to the AERONET site position. For the MISR V22 validation, the sampling area is $\sim 50 \times 50$ km$^2$ (9 MISR pixels) and the temporal window is $\pm 1$ h (Kahn et al., 2010), while Garay et al. (2017) use the single pixel and closest time approach for validation of the 4.4 km MISR aerosol product. For AATSR AOD validation a spatial window of $\pm 35$ km and temporal window of $\pm 30$ min is typically used (de Leeuw et al., 2015; Popp et al., 2016). Petrenko et al. (2012) found that the difference in using a circular area with 50 km radius or a squared area with 50 km side ($5 \times 5$ pixels) for MODIS validation did not have a large effect, except that for the

circular area there are 22% fewer data points. Here we use a circular area around the AERONET site for sampling the satellite data and a range of sampling parameters that covers the previously used values.

In this paper we study the effect of the sampling parameters, the sampling distance $d$ for averaging the AATSR data around an AERONET site and the time window $\Delta t$ for sampling the AERONET data, on the comparison of AATSR and AERONET DRAGON campaign AOD data. We use ten sampling distances ranging from $0.05°$ to $1.0°$, where the smallest $d$ corresponds to a single AATSR pixel coinciding with the AERONET site location, and the largest $d$ corresponds to sampling almost the entire test area. For the temporal sampling we use six values for $\Delta t$ ranging from $\pm 0.1$ h to $\pm 2$ h. A typical temporal sampling rate of the AERONET data is 15 minutes, so the smallest $\Delta t$ corresponds to a single observation closest in time to the satellite overpass. It is noted that some AERONET sites use a more frequent sampling rate, but we assume that this has a negligible effect, and have not differentiated these sites. The number of observations for both spatial and temporal sampling windows and the associated standard deviation are recorded.

A simple measure of the representativeness of the point-like AERONET observation for the larger area covered by the AATSR data is obtained from the standard deviation of the AATSR AOD ($\sigma_{\mathrm{AATSR}}$) around the AERONET site. For highly varying AOD the point-like measurement is likely to be less representative. In this sense $\sigma_{\mathrm{AATSR}}$ serves as a quantitative measure of the collocation mismatch uncertainty. It must be noted, however, that the variation of the AATSR AOD values around a site is not necessarily due to the natural variability of aerosol loads alone, but is likely affected by ADV retrieval errors. Hence $\sigma_{\mathrm{AATSR}}$ is not a direct measure of the collocation mismatch uncertainty. The dominant error sources in the satellite aerosol retrieval are residual clouds and varying surface reflectance in connection with the satellite dual view collocation uncertainties.

In this paper we use the AERONET DRAGON campaign data to assess the spatial variability of AOD on a scale similar to the AATSR AOD L1 product grid ($\sim 10$ km). Similar to the AATSR sampling, for each AERONET DRAGON site we calculate the average AOD of the nearby AERONET sites (within the sampling distance) and the corresponding standard deviation of AOD ($\sigma_{\mathrm{AERO}}^{\mathrm{NEAR}}$). We can then compare this to the corresponding $\sigma_{\mathrm{AATSR}}$ for each match between AATSR and AERONET during the DRAGON campaign. We also calculate the temporal standard deviation of AERONET AOD from the observations within the temporal sampling window for each AERONET site ($\sigma_{\mathrm{AERO}}$).

The number of retrieved satellite pixels in the sampling area around an AERONET site ($N_{\mathrm{AATSR}}$) gives a simple measure of the sampling uncertainty. $N_{\mathrm{AATSR}}$ is mainly affected by cloud screening, and a large number of clouded pixels may imply an elevated probability of residual clouds, and thus low $N_{\mathrm{AATSR}}$ indicates higher sampling uncertainty. The number of nearby AERONET sites ($N_{\mathrm{NEAR}}$) used when calculating $\sigma_{\mathrm{AERO}}^{\mathrm{NEAR}}$ is also recorded, and can be used as a threshold. A third number associated in sampling the data is the number of temporal samples for the AERONET site ($N_{\mathrm{AERO}}$). A low number of samples indicates weaker statistics in calculating the standard deviations. The $0.1°$ resolution pixels used in the standard ADV AOD retrievals consist of approximately 100 subpixels in the nominal 1 km resolution of AATSR. A representative sample of the subpixels is selected for calculating the top of atmosphere (TOA) reflectance for the $0.1°$ retrieval area, and the standard deviation of TOA reflectance $\sigma_{\mathrm{RTOA}}$ at 555 nm is recorded for quality assurance (Kolmonen et al., 2016). High variability in the measured TOA reflectance for a retrieval area may indicate residual clouds or variable surface reflectance, which are considered the major sources of uncertainty in the satellite aerosol retrievals.

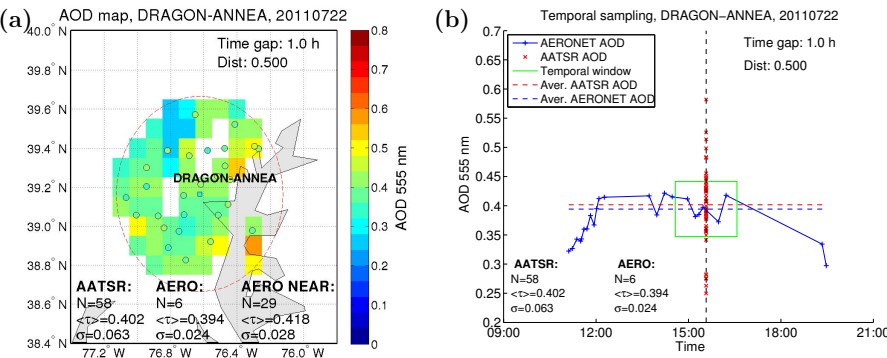

**Figure 2. (a)** Spatial sampling in the AOD comparison. The dashed red circle shows the sampling area around an AERONET site (DRAGON_ANNEA, colored square). The colored boxes show the AOD values for AATSR pixels for the example scene (22 July 2011). The colored symbols show the AOD values of the other AERONET sites within the sampling area. The text inset at the bottom of the image shows the number of samples, average AOD, and the standard deviation for AATSR, the central AERONET site, and for the nearby AERONET sites. **(b)** Temporal sampling in the AOD comparison. The green rectangle indicates the temporal sampling window. The blue line and symbols show the AERONET observations for the selected site as function of time (UTC), the red symbols show the AATSR AOD values at the overpass time. The dashed horizontal lines show the averaged AOD values for AATSR (red) and AERONET (blue) using the selected sampling parameters ($d = 0.5°$, $\Delta t = 1$ h). See Sect. 3.2 for more details.

Figure 2 illustrates the sampling used in the satellite AOD validation. The sampling distance $d$ defines the radius of the circular area around an AERONET site used for the spatial sampling of the AATSR data. The temporal sampling parameter $\Delta t$ defines the time window used for averaging the AERONET data. In this example from 22 July 2011, the $N_{\text{AATSR}}$=58 AATSR pixels within the sampling area give an average AOD of 0.40 with a standard deviation $\sigma_{\text{AATSR}}$=0.06, while the $N_{\text{AERO}}$=6 temporal samples at the AERONET site DRAGON-ANNEA give a temporal AOD average of 0.39 and a temporal standard deviation $\sigma_{\text{AERO}}$=0.02. The spatial sampling area is also used to study the spatial variability of the AERONET data by considering the nearby AERONET sites within the sampling distance. The AOD values from the $N_{\text{NEAR}}$=29 nearby sites are first averaged temporally for each site respectively, and then spatially to get the spatial average AOD of 0.42 and corresponding spatial standard deviation of $\sigma_{\text{AERO}}^{\text{NEAR}}$=0.03.

## 4  Results

### 4.1  AOD comparison

Figure 3 (a) shows the basic AOD comparison between the spatially and temporally collocated AATSR ADV and AERONET results for the DRAGON campaign. The AOD comparison with a sampling distance of $d = 0.2°$ and sampling time window $\Delta t = 0.25$ h shows a decent agreement with a correlation coefficient of $R = 0.94$ for the 210 collocated matches between AATSR and AERONET. The average AOD values agree at $\sim 0.2$ with a slight high bias for AATSR, and there are some

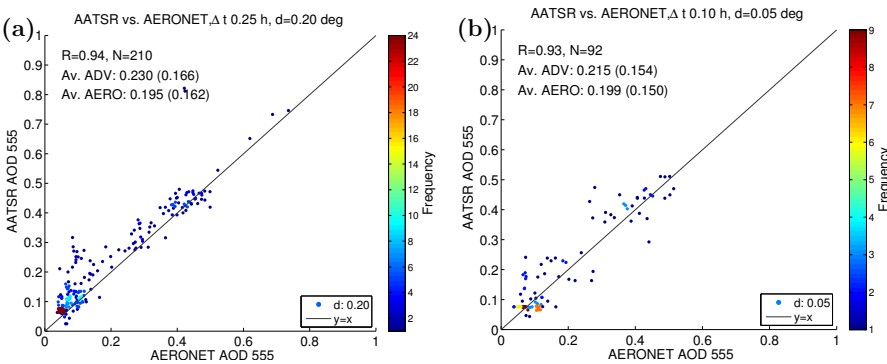

**Figure 3. (a)** Comparison of AATSR and AERONET AOD values for the 2011 DRAGON campaign, using a sampling distance $d = 0.2°$ and a time window $\Delta t = 0.25$ h. **(b)** Comparison with $d = 0.05°$ and a time window $\Delta t = 0.1$ h, corresponding to closest point comparison. The text insets show the correlation coefficient $R$, number of matches $N$, and average AOD values (standard deviations).

outliers at larger AOD values. Panel (b) shows the comparison using the smallest sampling parameters, corresponding to a single satellite pixel that encloses the site and a single ground based observation closest in time to the satellite overpass. We see that the correlation coefficient for AOD is equally good as in panel (a), but the number of matches is significantly reduced and the data is more scattered.

In Fig. 4 the effect of sampling parameters on the AOD comparison is shown more systematically. In panel (a) we plot the correlation coefficient $R$ between the collocated AOD values as a function of the sampling distance $d$ for several temporal sampling windows $\Delta t$. We see that for the smallest $d$ the correlation is poor, except for the smallest $\Delta t$. The peak correlation is obtained at $d = 0.4 - 0.6°$, after which the correlation decreases. The correlation is weaker for larger $\Delta t$.

Figure 4 (b) shows the average AOD of the collocated matches for both AATSR and AERONET. It is emphasized here that these averages are calculated from the collocated matches only (not the full data sets), and the set of matches depends on the sampling parameters. Thus the sampling distance has an effect on the average AERONET AOD, even though the AERONET sampling has no direct dependence on $d$. Similarly, the average AATSR AOD depends indirectly on $\Delta t$. It must also be noted that the same AATSR pixels may contribute to the samples corresponding to several AERONET sites. We see that the average AOD increases with increasing $\Delta t$. This can be understood as a cloud proximity effect: some of the potential matches between AATSR and AERONET are removed because of cloud screening by the AERONET algorithm, when smaller $\Delta t$ are used. When the sampling parameters are relaxed, the additional matches so obtained are more likely to include observations made in the proximity of clouds. These matches have enhanced AOD due to either cloud contamination (cloud affected pixels interpreted as clear sky AOD) or actual enhancement of AOD in the proximity of clouds, due to e.g. hygroscopic growth. The enhancement of AERONET AOD in the proximity of clouds has been studied e.g. by Eck et al. (2014), Arola et al. (2017), and others. We note that for the largest $\Delta t$ the AERONET data may be affected by the diurnal effects (Kaufmann et al., 2000; Smirnov et al., 2002; Arola et al., 2013), but this does not explain the increase in the average AATSR AOD. We also notice that the average AOD decreases slightly with increasing sampling distance. This cannot be explained by cloud contamination. Panel

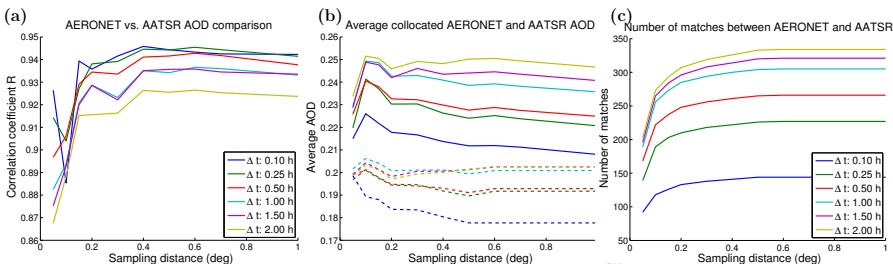

**Figure 4. (a)** Dependence of the AOD correlation coefficient $R$ on the AATSR sampling distance $d$ and AERONET sampling time window $\Delta t$. **(b)** Dependence of the average AOD of the sampled data on $d$ and $\Delta t$. The solid lines show the AATSR values and the dashed lines of the corresponding color show the AERONET values. The line colors are the same as in panel (a). **(c)** The number of matches between AATSR and AERONET as function of $d$ and $\Delta t$.

(c) shows how the number of matches between AATSR and AERONET increases with the sampling distance and temporal window size, reaching N=334 at highest. The plotted values are summarized in Table S1 (in the supplement).

## 4.2 AOD variability comparison

The comparison of AOD variability, as measured by the spatial standard deviation of AOD within the sampling area ($\sigma_{\mathrm{AOD}}$) for AATSR and AERONET, respectively, shows much less agreement with $R = 0.49$ for a sampling distance of $d = 0.5°$, and even less for $d = 0.2°$ (Fig. 5). Here we have required at least three samples from both data sources for calculating the standard deviations. The AATSR AOD variability ($\sigma_{\mathrm{AATSR}}$) is much larger on average than the corresponding AERONET value ($\sigma_{\mathrm{AERO}}^{\mathrm{NEAR}}$), and there are a lot of outliers in the scatter plot. Increasing the sampling distance improves the correlation, but many outliers remain. The effect of sampling distance on the AOD variability comparison is shown systematically in Fig. 6. The average AOD variability (over the collocated matches) increases steeply as $d$ is increased, until it starts to saturate at $d > 0.5°$. The average $\sigma_{\mathrm{AATSR}}$ is often more than twice that of the corresponding $\sigma_{\mathrm{AERO}}^{\mathrm{NEAR}}$, but the dependence on $d$ is similar. The larger variability of aerosol optical depth for the satellite data indicates that the noise or retrieval errors in the satellite data affect the variability estimate considerably. In Fig. 6 (b) we see that the correlation coefficient $R_\sigma$ between $\sigma_{\mathrm{AATSR}}$ and $\sigma_{\mathrm{AERO}}^{\mathrm{NEAR}}$ is quite low for $d = 0.15 - 0.4°$, but increases with increasing sampling distance. For the smallest $d$ there is a lot of variation due to the low number of matches. We assume that the random noise in the satellite data is averaged out when the sampling distance is increased. The actual aerosol variability is then better exposed, leading to improved correlation with the ground based data. There also seems to be a systematic component leading to the high bias. The temporal sampling parameter $\Delta t$ does not have so large effect on $R_\sigma$ and the dependence on it is not very systematic, but the smallest $\Delta t$ typically give the worst correlation. The results of the comparisons with different sampling radii are summarized in Table S2.

The dependence of the temporal variability in the AERONET AOD data on the sampling parameters is shown in Fig. S1 (a) in the supplement. As expected, the dependence on $d$ is weak, but $\Delta t$ has a considerable effect. Figure S1 (b) shows the correlation coefficient between the spatial variability of AATSR AOD and the temporal variability of AERONET AOD for

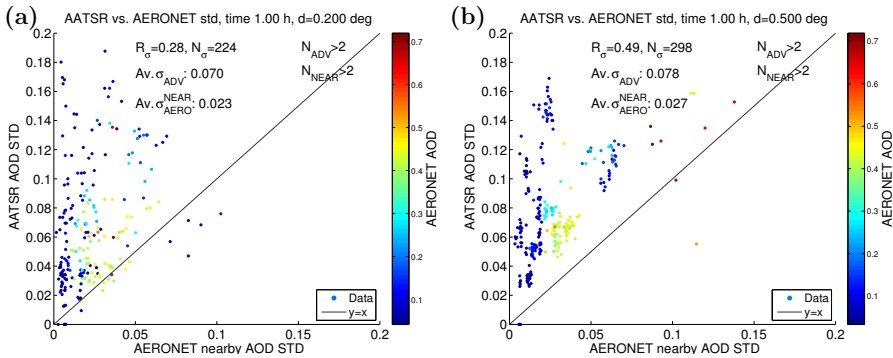

**Figure 5.** Comparison of spatial aerosol variability estimates between AATSR and AERONET, calculated from the standard deviation of AOD within the sampling distance, for $d = 0.2°$ (**a**) and $d = 0.5°$ (**b**). The temporal sampling window used here is $\pm 1$ h, and at least three observations in the spatial sampling area are required for each data source ($N_{\mathrm{ADV}} > 2$ and $N_{\mathrm{NEAR}} > 2$). The color shows AERONET AOD at 555 nm. Higher AOD values are generally associated with higher variability.

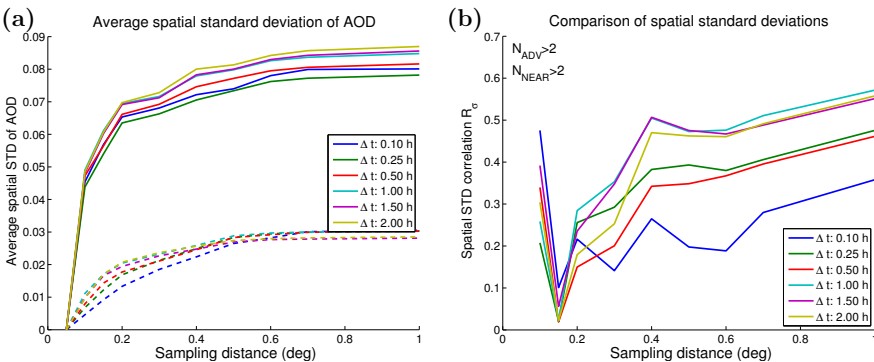

**Figure 6.** (**a**) Average spatial standard deviation of AOD for AATSR (solid lines) and for AERONET (dashed lines) for the collocated cases as function of the sampling distance for several temporal sampling window sizes. (**b**) The correlation coefficient $R_\sigma$ for collocated spatial AOD variability data as function of the sampling distance.

various sampling parameter values. The correlation improves with increasing sampling distance, and is typically highest for $\Delta t$=0.5 h. On average, some of the air mass sampled by the satellite at the overpass time is also sampled by the AERONET instruments in the given time window. The fraction of the mutually sampled airmass depends on the wind speed and the size of the sampling windows, explaining the variation seen in Fig. S1 (b). Figure S1 (c) shows a similar comparison, but with the spatial AOD variability obtained from the nearby AERONET sites. The correlation between the spatial and temporal variability is then generally higher than when using the AATSR data. Here we have required that the number of samples for both AATSR and AERONET is at least 3 when calculating the standard deviations. We have also removed cases where the number of matches is low.

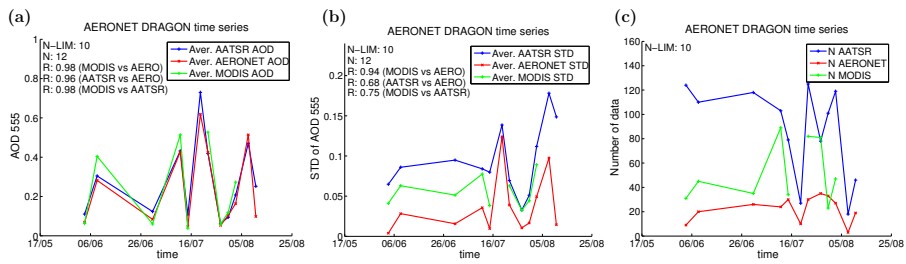

**Figure 7. (a)** Time series of AOD for the entire DRAGON campaign area for AATSR ADV, AERONET, and MODIS. The AERONET data has been temporally averaged in a 1 h time-window centred at the AATSR overpass time for each day. Overpasses with less than 10 AATSR or MODIS data points in the area, respectively, are excluded. The AERONET and MODIS data are shown only for the days when AATSR data is available. The text inset shows the AOD correlation coefficients for three cases: MODIS against AERONET, AATSR against AERONET, and MODIS against AATSR. **(b)** Corresponding plot for the spatial standard deviation $\sigma_{\mathrm{AOD}}$. **(c)** The number of AATSR and MODIS pixels in the area for each day, and the number of AERONET sites with data close to the satellite overpass time ($\pm 0.5$ h).

Note that in studying the aerosol variability some of the data are 'double counted': the sampling areas for nearby AERONET sites overlap, and thus the AOD value for a particular location is used several times in the comparison, for both AATSR and AERONET. To avoid this, we have also made a comparison using the whole DRAGON campaign area, i.e. for each satellite overpass we calculate the average AOD and the corresponding standard deviation for the whole area, without spatially collocating the individual sites and pixels. A temporal collocation with a $\pm 0.5$ h sampling window for the AERONET data is used. From the 21 AATSR overpasses during the campaign we have removed 9 days with a limited number of data points (less than 10), when the satellite orbit only partly overlaps with the study area or the scene is heavily clouded. The time series constructed in this way is shown in Fig. 7. In general there is agreement between AATSR and AERONET for the area as whole, both for AOD and the spatial standard deviation of AOD. $\sigma_{\mathrm{AATSR}}$ is systematically larger than $\sigma_{\mathrm{AERO}}^{\mathrm{NEAR}}$, but they change in the same manner. Large differences in AOD are associated with large standard deviation in the retrieved AATSR AOD, indicating high aerosol spatial variability (or large retrieval errors). We have also included in the plots the MODIS 10 km AOD product data averaged over the study area, obtained from the near-simultaneous Terra orbits, for reference. The use of MODIS data is discussed in section 4.3. The values plotted in Fig. 7 are summarized in Table S3.

It is noted that the use of a circular sampling area around the sites may not always be the optimal choice for the comparison, since the AOD variability is not necessarily symmetric due to the effects of local topography on the aerosol transport. In addition, the surface reflectance variability around each AERONET site may systematically affect the satellite retrievals. To reveal possible issues with any of the AERONET sites, we consider the results over the campaign period respectively for each site in Fig. S2 and Table S4 in the supplement. The CMU and AOD correlation vary from site to site, but none of the sites stands out as particularly peculiar, when the full range of sampling parameters is considered. We note that the low number of matches for individual sites limits the analysis; more data would be required to study the CMU in greater detail for individual

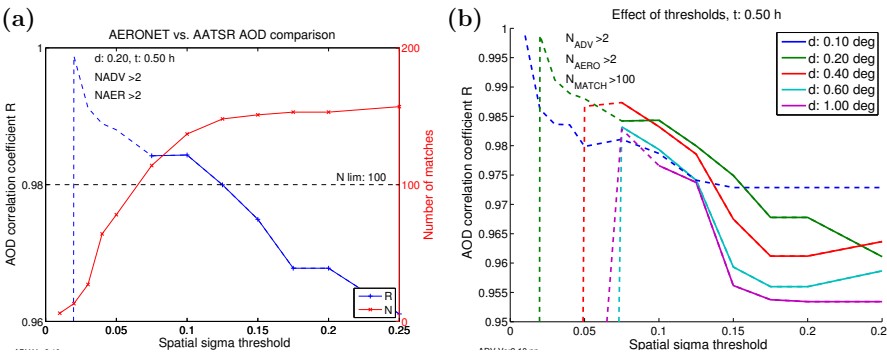

**Figure 8. (a)** Effect of $\sigma_{\mathrm{AATSR}}$ threshold on the AOD comparison with $d = 0.2°$ and $\Delta t = 0.5$ h. We have also applied the thresholds $N_{\mathrm{ADV}} > 2$ and $N_{\mathrm{AERO}} > 2$ to ensure sufficient statistics for calculating $\sigma_{\mathrm{AATSR}}$. The blue line shows the AOD correlation coefficient $R$ (left y-axis), and the red line shows the corresponding number of matches $N$ (right y-axis) after an upper threshold on $\sigma_{\mathrm{AATSR}}$ (x-axis) has been applied. Results with $N < 100$ (dashed horizontal black line) are shown by the dashed blue line. **(b)** The same for various sampling distances. The dashed lines show results for which less than 100 matches are left.

sites. Furthermore, we did not find any systematic difference between urban and non-urban sites in the spatial (AATSR) or temporal (AERONET) standard deviation of AOD.

Next, we consider the effect of various thresholds applied to the data before the comparison. The primary parameter is $\sigma_{\mathrm{AATSR}}$, as we want to explore its usefulness as an estimate of the collocation mismatch uncertainty. The idea is that if $\sigma_{\mathrm{AATSR}}$ describes the collocation mismatch uncertainty (or the representativeness of the AERONET data in validating satellite AOD results), then the application of an upper threshold ($\sigma_{\mathrm{AATSR}}^{\mathrm{threshold}}$) to this parameter should improve the correlation between AATSR and AERONET data. Figure 8 shows that this is true for a certain range of $\sigma_{\mathrm{AATSR}}^{\mathrm{threshold}}$: the threshold starts to have effect when $\sigma_{\mathrm{AATSR}}^{\mathrm{threshold}} < 0.2$, and improves the AOD correlation until $\sigma_{\mathrm{AATSR}}^{\mathrm{threshold}} \sim 0.1$, which is close to the average $\sigma_{\mathrm{AATSR}}$ as seen in Fig. 6. After this, the number of matches is quickly reduced, and the effect of $\sigma_{\mathrm{AATSR}}^{\mathrm{threshold}}$ is dubious. We note that even though applying an upper threshold for $\sigma_{\mathrm{AATSR}}$ improves the AOD correlation, $\sigma_{\mathrm{AATSR}}$ may not describe the actual AOD variability (or collocation mismatch uncertainty), but is affected by retrieval errors. This is evident from Fig. S3, where similar thresholds applied to the AOD variability obtained from the AERONET data do not result in clear improvement in the correlation coefficients.

Other parameters related to the comparison statistics are $N_{\mathrm{ADV}}$, $N_{\mathrm{NEAR}}$, $\sigma_{\mathrm{AERO}}^{\mathrm{NEAR}}$, $N_{\mathrm{AERO}}$, $\sigma_{\mathrm{AERO}}$, and $\sigma_{\mathrm{RTOA}}$, as described in section 3.2. Figure S4 (a) shows the effect of $N_{\mathrm{ADV}}$, the number of AATSR pixels within the sampling area. $N_{\mathrm{ADV}}$ can be used as a measure of fractional cloud cover. Clouded pixels are removed in the algorithm, and a low $N_{\mathrm{ADV}}$ (with respect to a maximum value when all pixels are retrieved) indicates that clouds are present. A patchy cloud mask indicates elevated probability of cloud contamination and overestimated AOD. A lower threshold for $N_{\mathrm{ADV}}$ is also crucial when calculating $\sigma_{\mathrm{AATSR}}$ to ensure sufficient statistics. The same considerations apply to $N_{\mathrm{NEAR}}$, the number of nearby AERONET sites in the spatial sampling area (Fig. S4 (d)). In Fig. S4 we see that a more stringent threshold for $N_{\mathrm{ADV}}$ or $N_{\mathrm{NEAR}}$ improves the agreement between

AATSR and and AERONET in AOD comparison. Figure S4 show the effect of these thresholds in the the AOD variability comparisons.

In Fig. 8 we have required a minimum number of three AATSR samples ($N_{\mathrm{ADV}} > 2$) and three AERONET samples ($N_{\mathrm{NEAR}} > 2$) for each match when calculating the standard deviations. This is a rather low limit, and further improvement in the agreement can be obtained by applying more stringent thresholds as seen in Figs. S4 and S5. However, this does not apply to the smallest sampling parameters, for which the maximum number of samples is already very limited. Therefore we have used these moderate thresholds when comparing the spatial AOD variations. Figure S4 (c) shows the effect of applying thresholds for $\sigma_{\mathrm{RTOA}}$, the average subpixel standard deviation of the top of atmosphere (TOA) reflectance at 555 nm. Applying a $\sigma_{\mathrm{RTOA}}$ threshold improves the AOD comparison results slightly, although not systematically. For the AOD variability correlation coefficient $R_\sigma$ the improvement is more significant and more systematic, as seen in Fig. S5 (c).

The thresholds can be optimized to improve the correlation between AATSR and AERONET aerosol variability estimates. Table S7 shows that the $\sigma_{\mathrm{AOD}}$ correlation can be brought close to 0.7 by applying a suitable set of thresholds to the collocated, spatially averaged data. It is seen that removing the cases with a low number of AATSR and AERONET data improves the agreement, as well as removal of cases with high average $\sigma_{\mathrm{RTOA}}$. However, such threshold sets are usually case dependent, and further studies would be needed for other regions with different circumstances.

## 4.3   Comparison with MODIS

To further test our comparison approach, we apply similar analyses to MODIS Terra Collection 6 AOD data, i.e. we test the effect of sampling parameters on the comparison with AERONET. Munchak et al. (2013) compared MODIS AOD data from Terra and Aqua to AERONET in two approaches: single pixel comparison, and spatial averaging with 50 km radius for both 3 km and 10 km AOD products. We expand this approach by using a number of sampling lengths and sampling time windows (Fig. 9) as with AATSR. We also consider the AOD variability, which was not addressed by Munchak et al. (2013).

Figure S6 shows the comparison of MODIS AOD and $\sigma_{\mathrm{AOD}}$ against AERONET. We see that agreement between MODIS and AERONET is similar to that between AATSR and AERONET in terms of the correlation coefficients. However, in the AOD comparison there is a large systematic positive bias for MODIS, which is specific to Terra (Levy et al., 2013). For $\sigma_{\mathrm{AOD}}$ MODIS shows slightly better correlation with AERONET. The average AOD variability for MODIS is 0.05-0.06, for AATSR $\sim$0.08, and for AERONET $\sim$0.03. The 3 km data agrees less well both for AOD and for $\sigma_{\mathrm{AOD}}$, but has more matches with AERONET i.e. better coverage.

Figures 9 (a) and (c) show similarity with the AATSR results (Figs. 4 and 6): as with AATSR, the best agreement between MODIS and AERONET AOD observations is obtained with the smaller sampling distances ($d = 0.2^\circ - 0.4^\circ$), while the agreement for AOD variability increases with the sampling distance. Note that the y-axis scale is different in Figs. 4 (a) and 9 (a), since the correlation with AERONET at short distances is lower for AATSR. Figures 4 (b) and 9 (b) show that the average AOD is lowest for the smallest temporal sampling windows, but the dependence on the sampling distance is different. For MODIS, the average AOD decreases systematically with the sampling distance, which cannot be explained by a cloud proximity effect. Figure S7 shows the number of matches and the standard deviation of AOD as function of $d$, and the effect of a $\sigma_{\mathrm{AOD}}$ threshold

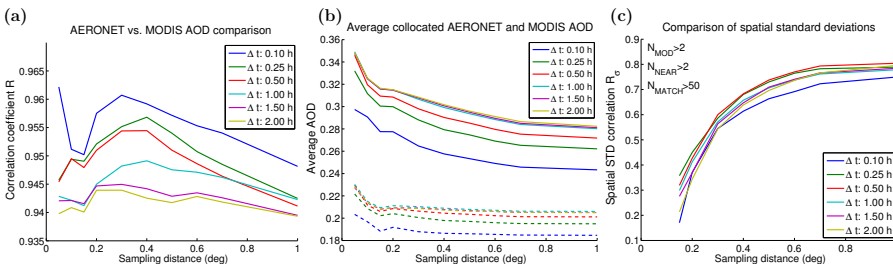

**Figure 9. (a)** Dependence of the AOD correlation $R$ between AERONET and MODIS 10 km product on the sampling distance. The colors indicate the temporal sampling window size. **(b)** Dependence of the average AOD for the matching cases on the sampling distance. on the sampling distance. The solid lines correspond to MODIS 10 km product and dashed lines correspond to AERONET data. **(c)** Dependence of the AOD spatial variability correlation for the matching cases on the sampling distance. Here we have required at least three observations from each data source to calculate $\sigma_{\mathrm{AOD}}$, and the correlation coefficient is not shown of the number of matches is less than 50.

for the MODIS 10 km product. These are largely similar to the AATSR results. In particular, setting an upper threshold for the spatial AOD variability calculated from the MODIS data improves the AOD correlation slightly. Figure S8 shows the same results for the MODIS 3 km product. There is more variation in the correlation coefficients for the 3 km product, but in general the dependencies on the sampling parameters are similar.

The day-to-day changes in AOD and spatial AOD variability for the full study area are tracked in a similar fashion by both AATSR and MODIS, as seen in Fig. 7. Note that in this figure we limit the consideration to the days when AATSR has data over the area; for MODIS and AERONET there is data for more days than shown here. We note that MODIS AOD data is missing for the days when the AOD and AOD variability are the highest, possibly due to more stringent cloud screening than used for AATSR. MODIS overestimates the variability less than AATSR and has slightly better correlation with AERONET. To further compare the AOD results from the two satellite instruments, we have regridded the MODIS 10 km AOD product data to the $0.1°$ AATSR grid for near-simultaneous retrievals in the study area. We find that the collocated AOD values agree well with R=0.89 (Fig. S10), with MODIS AOD exceeding that of AATSR by 0.06 on average. We note that to properly compare the spatial variability of AOD between the two satellite instruments would require more careful sampling, which is beyond the scope of this study.

## 4.4  Total uncertainty

The uncertainty estimates related to satellite aerosol products are increasingly researched, as they are crucial e.g. in assimilating the satellite data to models. For example, the three main AATSR algorithms all provide per pixel uncertainty estimates for AOD (de Leeuw et al., 2015; Popp et al., 2016). Here we consider the additional value of the collocation mismatch uncertainty estimate, as obtained from the spatial standard deviation of the satellite AOD, in the AOD validation. Again, we concentrate on the AATSR Dual View (ADV) algorithm and AERONET data from the DRAGON 2011 campaign. We compare the AOD retrieval error ($\Delta\tau = |\tau_{\mathrm{AATSR}} - \tau_{\mathrm{AERO}}|$) to the uncertainty estimate in two steps. First, we use only the standard AOD uncer-

tainty estimate as obtained from ADV, which is based on the observation uncertainty propagated through the retrieval process (instrument uncertainty). Next, we consider the total uncertainty including the collocation mismatch uncertainty, estimated by the AATSR AOD standard deviation within the sampling area used around each AERONET site. In the total uncertainty, the AERONET AOD data is considered as reference data (the 'ground truth'), with a systematic AOD uncertainty of 0.01 (Eck et al., 1999).

In order to take the uncertainties into account in the AOD validation, we need to use other metrics in addition to the correlation coefficient, root means square error, and the linear regression parameters. Adopting the approach of Immler et al. (2010), we consider the equation

$$|m_1 - m_2| \leq k\sqrt{u_1^2 + u_2^2 + \sigma^2}, \tag{1}$$

where $m_i$ are the measured values (by AATSR and AERONET, respectively), $u_i$ are the corresponding uncertainties, and $\sigma$ corresponds to the collocation mismatch uncertainty. The factor $k$ is the so-called coverage factor, which describes the consistency of the data. In the terminology proposed by Immler et al. (2010), when Eq. (1) holds for $k = 1$ the data are 'consistent', and the data are 'in agreement' if the equation holds for $k = 2$. If the equation does not hold even for $k = 3$ the data are 'inconsistent'. Figure 10 (a) shows how Eq.(1) holds for the data without CMU. The 'AOD uncertainty' here is $\sqrt{u_1^2 + u_2^2 + \sigma^2}$, where $u_{\text{AERO}}$ is fixed at 0.01 and $\sigma = 0$. The colored lines correspond to $k = 2$ (red) and $k = 3$ (cyan), while the dashed black line corresponds to $k = 1$. We see that for most of the points (92 %) the 'data are consistent' i.e. below the $k = 1$ line, and there are no points above the $k = 3$ line (inconsistent data). In Fig. 10 (b) we have included the CMU ($\sigma = \sigma_{\text{AATSR}}$). The fraction of the 'consistent' pixels is then increased to 98 %, at the cost of a 19 % increase in the average uncertainty. We see that inclusion of CMU also improves the correlation coefficient $R$ between the AOD difference and the uncertainty, indicating that the CMU is larger for the cases with larger error.

Unlike the AATSR algorithms, MODIS does not provide per pixel uncertainty estimates. Instead, expected error values, based on global validation results, are provided. For MODIS Collection 6 over land 69.4% of data fall within $\pm 0.05$ or $\pm 0.15 \times$AOD from the true value (Levy et al., 2013). Figure S9 shows the scatter plot of MODIS 'uncertainty' against the AOD error (difference to AERONET). Here we assume that the MODIS AOD uncertainty consists of a constant part 0.05 and an AOD dependent part, 0.15 times the AOD. Figures S9 (b) and S9 (d) show the effect of adding the collocation mismatch uncertainty obtained from the standard deviation of the MODIS AOD within the sampling area; this increases the fraction of consistent pixels from 55% to 61% for the 10 km product, and from 42% to 57% for the 3 km product. Hence the AOD variability estimates might be useful also for the MODIS uncertainty budget.

## 5    Conclusions

Three main conclusions can be made.

1) The results of a satellite AOD validation against AERONET data depend on the sampling parameters used in the validation due to the AOD variability. For both MODIS and AATSR data there is an 'optimal' sampling radius of $\sim$0.3-0.4°, which gives

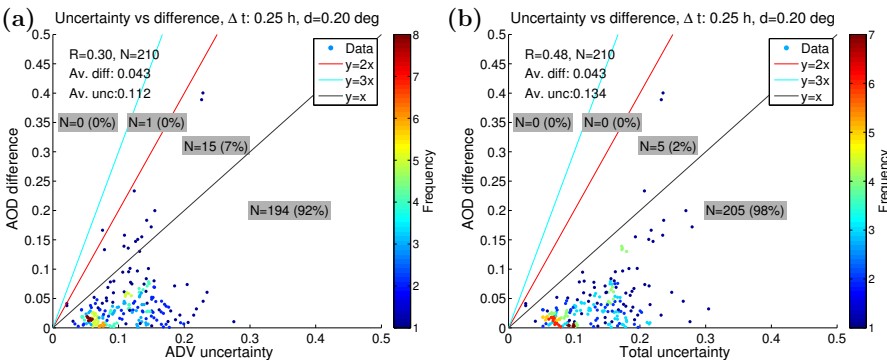

**Figure 10.** Scatter plot of error vs. uncertainty illustrating Eq. (1). The text inset at top left show the correlation coefficient $R$ between the AOD difference ($|\tau_{\text{AATSR}} - \tau_{\text{AERO}}|$) and uncertainty estimate, the number of matches $N$, and the average difference and average uncertainty. The colored lines correspond to different values of the coverage factor $k$ (see text). The text insets on dark background indicate the number (fraction) of pixels within each 'consistency class'. **(a)** Uncertainty due to ADV AOD uncertainty only. **b**) Total uncertainty including collocation mismatch uncertainty estimate.

the best correlation coefficient. The correlation decreases when the sampling distance increases further, as the AOD variability starts to have a larger role. The temporal sampling has a less significant but non-negligible effect. Best correlation is obtained with the shortest sampling time window. The average AOD over all matches between the satellite and AERONET data depends on the sampling parameters. The dependence is different for AATSR and MODIS, and requires further investigation.

2) We find that the local AOD variability obtained from satellites and from the ground based data correlate only weakly for short sampling distances. The satellite based AOD variability can be several times larger than its ground based counterpart, apparently due to noise caused by retrieval errors. The correlation can be increased by using larger sampling area size, which smooths the random noise in the satellite data. On a day-to-day basis, the satellite derived $\sigma_{\text{AOD}}$ values for larger area follow the relative changes observed in the AERONET data well, while the absolute values are high. The number of data within each sampling windows are important quality parameters in the validation.

3) The correlation mismatch uncertainty estimate obtained from the standard deviation of satellite AOD has some use in describing the validation results. If an upper threshold is applied on the satellite AOD variation in the sampling area around an AERONET site, the correlation between the collocated satellite and ground based AOD values is slightly improved. From another point of view, if the collocation mismatch uncertainty estimate is taken into account when comparing the retrieval error and total uncertainty, the fraction of consistent measurements is increased.

*Data availability.* A collocated AATSR ADV v2.30 and AERONET dataset with varying sampling parameters is available via the GAIA-CLIM project at ftp://ftp-ae.oma.be/dist/GAIA-CLIM/D3_6/AOD/FMI/'. The ADV v3.10 data is available from the ICARE web service, http://www.icare.univ-lille.fr.

*Competing interests.* The authors declare that they have no conflict of interest.

*Acknowledgements.* We thank the AERONET Principal Investigators and their staff for establishing and maintaining the sites used in this investigation. We thank the researchers involved in the NASA's DISCOVER-AQ campaign and in the AERONET DRAGON 2011 campaign. This work was supported by the Horizon 2020 grant GAIA-CLIM, Grant no. 640276. This work was supported by Academy of Finland Finnish Centre of Excellence, Grant no. 272041.

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
