# Peer review of "Collocation Mismatch Uncertainties in Satellite Aerosol Retrieval Validation"

_Atmospheric Measurement Techniques, 2017_

## Referee Comment (RC1) · Anonymous Referee #1 · 17 Dec 2017

This paper systematically investigates the effect of spatial and temporal collocation windows between satellite and ground observations on the evaluation of satellite AOD. It shows that spatial and temporal variability exhibited in AOD may exert significant impact on the comparison results, and accounting for the collocation mismatch uncertainty will improve the agreement. This topical is critical in satellite data validation and the finding of the current study provides important reference and guidance for future satellite-ground intercomparisons. The paper is also well written and easy to follow. I only have a few minor comments:

Specific comments:

1. The paper focuses on the DRAGON campaign area and mostly provides composite results for all sites. I wonder if how different is the CMU for different sites, especially

between urban and rural sites?

2. The "collocated area" between satellite and ground measurements is defined as a symmetric circle around the AERONET site. I understand this is conventional practice for satellite data evaluation. But due to factors such as aerosol transport, cloud and topography, etc, the distribution of AOD spatial variability is usually not symmetric. I wonder if the authors have examined the specific AOD spatial variability for each site?

3. In this work and in all collocation works the CMU actually combines both spatial and temporal variability, as both a space and time window is needed. It would be more precise if the space and time CMU could be separated. I understand this is a difficult practice, but could the authors offer some discussion?

4. The results indicate some difference between the CMU estimated using AATSR and MODIS data. Since AATSR is not as popular a dataset as MODIS. It would be helpful to offer some intercomparison results between these two datasets, e.g., any disagreements in their absolute magnitudes and spatial variability.

5. Figures 4, 6, and 7-9 seem a bit difficult to read. I suggest increase the line weights and fond sizes a little bit for clearer presentation.

---

## Referee Comment (RC2) · Anonymous Referee #2 · 22 Dec 2017

General comments:

The study compares satellite-based data from AATSR and MODIS to intensive ground-based data from the DRAGON phase in 2011 of the larger DISCOVER-AQ campaign, investigating the sensitivity of the comparison to the sampling parameters used to make collocations. Defining satellite data to be collocated if it falls within a spatial radius and temporal interval near a given ground site, the study found a sweet spot in which the sample is large enough to overcome random noise while remaining representative of the point data. All efforts to use ground-based sensors to validate satellite observations rely on similar parameter decisions, so the systematic approach in this study is helpful to a broad audience. The paper is well organized and easy to follow. Here are my specific comments.

[Figure]

Specific comments:

1. Figures in general have small text, which makes the legends difficult to read at the size they appear in the manuscript. Consider increasing the font sizes.

2. Page 6, lines 20-33. How does the range of the parameters used for previous work compare to the range used in this study?

3. Page 10, lines 6-7. Minimum sample number is another parameter that affects collocations. Do you know whether it's sensitive to the number of satellite pixels needed to form a match?

4. Page 14, line 11. The systematic positive bias in the MODIS data is specific to Terra, and discussed in Levy et al. (2013).

---

## Referee Comment (RC3) · Anonymous Referee #3 · 27 Dec 2017

This study presents extensive validation of AATSR AOD against AERONET observations made during the DRAGON 2011 campaign, which was held in the Baltimore-Washington metropolitan area. Comparisons between AATSR-retrieved and AERONET-observed AOD are performed as functions of various spatiotemporal sampling parameters and for different thresholds. The authors found that the spatial variability of AATSR (and possibly other similar sensors) AOD can be used as proxy of collocation mismatch uncertainty (CMU) to better characterize the uncertainty of satellite-retrieved AOD. Although some portion of the study (changes in comparison statistics as functions of spatiotemporal sampling parameters) has already been published elsewhere, the use of densely distributed DRAGON campaign data allows a new perspective to look at the issue, so I recommend this paper for publication with minor revision.

[Figure]

General comment

The DRAGON data set allows quantification of 'real' spatial and temporal variabilities of AOD, thus quantification of 'real' CMU (as opposed to the simple standard deviation of AOD shown in this paper). I believe that adding discussions on the real CMU and potential limitation of the satellite-derived CMU will improve the quality of the paper.

Specific comments

Page 1, line 13: the local AOD variability values correlate only weakly for short distance → the spatial variability correlates only weakly with that of AERONET for short distances

Page 1, line 18: the total uncertainty estimates → the total uncertainty estimates including the CMU

Page 2, line 14: based on the them → based on them

Page 3, line 10: and conclude that → and concluded that

Page 3, line 21 and some other places: on the average → on average

Page 7, lines 12-19: It is recommended to move this paragraph to the end of the section and modify to reflect connections between the statistics of this paragraph (number of data points, mean, standard deviation, etc.) and those of the following paragraphs (N-AATSR, N-AERO, sigma-AATSR, sigma-AERO, etc.).

Page 9, line 4 and other places: effect . . . to → effect . . . on

Page 16, line 9: sampling distance increases → sampling distance increases further

---

## Referee Comment (RC4) · Anonymous Referee #4 · 28 Dec 2017

The manuscript entitled "Collocation mismatch uncertainties in satellite aerosol retrieval validation" by Virtanen et al. investigates the detail of satellite retrieval uncertainties based on dense ground sun photometers network dataset. The authors well organized the topic so it should be published as the AMT paper. Note that a typo word "investigaged" in line 18 on page 2 should be corrected.

---

## Short Comment (SC1) · 11 Jan 2018

This is an interesting study that should have consequences for future satellite valida­tion work. The authors conclude that there is an optimal sampling distance of 0.3-0.4 degrees for such validations. That should give some confidence in 1 by 1 degree L3 products. I wonder though why correlations drop off much faster for MODIS data (Fig 9) than for AATSR data (Fig 4)? According to the authors, this drop-off is due to the natural variability in an AOD field (i.e. not related to retrieval errors). Wouldn't you expect MODIS and AATSR data to show similar results?

We recently studied the representativity of observations using high resolution mod­els: https://www.atmos-chem-phys.net/17/9761/2017/ . One interesting aspect is that

modelled fields show high correlation among neighbouring points while your satellite-AERONET comparison shows very low correlation amongst neighbouring points (i.e. short sampling distance). As the authors themselves conclude this points to large random errors in the satellite product that average out when using larger sampling distances.

---

## Author Comment (AC1) · 11 Jan 2018

**Response to Anonymous Referee #1**

**Collocation Mismatch Uncertainties in Satellite Aerosol Retrieval Validation**

Timo H. Virtanen, Pekka Kolmonen, Larisa Sogacheva, Edith Rodríguez, Giulia Saponaro, and Gerrit de Leeuw
Finnish Meteorological Institute, Helsinki, Finland

We thank the Referee for the positive feedback and constructive comments. The numbered comments are answered below and the changes to the manuscript are indicated in a separate pdf file.

1. *The paper focuses on the DRAGON campaign area and mostly provides composite results for all sites. I wonder if how different is the CMU for different sites, especially between urban and rural sites?*

   It was briefly mentioned in the manuscript (p. 5, line 4) that unlike Munchak et al. (2013), we did not find any significant difference in the performance of the satellite AOD retrieval between urban and rural areas. The same is true for CMU; we did not find any systematic difference between urban and non-urban sites in the spatial (AATSR) or temporal (AERONET) standard deviation of AOD. We went through the collocated cases (matches) for each site individually (Fig. 1 and Table 1 below). The CMU and AOD correlation vary from site to site, but none of the sites stands out as particularly peculiar, when the full range of sampling parameters is considered. We note that the low number of matches for individual sites limits the analysis; more data would be required to study the CMU in greater detail for individual sites. We have added the enclosed Fig. 1 and Table 1 to the Supplement (as Fig. S2, and Table S4), and a brief discussion of this to the manuscript (p. 12, after line 11).

[Figure]

Figure 1: Comparison of AOD and standard deviation of AOD between AATSR and AERONET does not differ systematically between urban and rural sites. The urban sites are marked with red circles. **(a)** Average AATSR and AERONET AOD for individual sites over the study period. **(b)** The correlation coefficient R between AATSR and AERONET AOD for individual sites. For one of the sites (DRAGON_Padonia) the AOD correlation coefficient is particularly low (0.6) for the selected sampling parameters (d=0.2°, $\Delta t = 1$ h), but not necessarily for other sampling parameters. **(c)** The average spatial standard deviation of AOD for AATSR and the corresponding temporal standard deviation of AERONET AOD for individual sites.

2. *The 'collocated area' between satellite and ground measurements is defined as a symmetric circle around the AERONET site. I understand this is conventional practice for satellite data evaluation. But due to factors such as aerosol transport, cloud and topography, etc, the distribution of AOD spatial variability is usually not symmetric. I wonder if the authors have examined the specific AOD spatial variability for each site?*

   It is understood that the spatiotemporal mismatch between the airmass sampled by the satellite and by the sun-photometers depends on the trajectories, i.e. wind direction and local topography. Although the use of AOD gradients (following Ichoku et al. 2002) was initially considered, we decided to focus in this study on CMU estimation methods that could be easily implemented in the global aerosol retrieval algorithm without the need for external data (such as wind directions). For the individual sites, see the response to the previous comment above. We have added a brief discussion of the matter to the manuscript (p. 12, after line 11).

   The effect of clouds on the collocation mismatch can be large even though both satellite and ground based data are cloud screened, as discussed in the manuscript (p. 9). A more detailed study taking into account the cloud proximity effects would be interesting, but is beyond the scope of this study.

3. *In this work and in all collocation works the CMU actually combines both spatial and temporal variability, as both a space and time window is needed. It would be more precise if the space and time CMU could be separated. I understand this is a difficult practice, but could the authors offer some discussion?*

In this study we concentrate mainly on the spatial AOD variability, since that is available from the satellite data and the goal is to provide the uncertainty estimate for the global satellite product. However, information on the temporal variability is available from the AERONET observations, as discussed in the manuscript (p. 7, line 10, and p. 10, line 20). The temporal variability and the correlation between spatial and temporal variability is shown in Fig. S1 in the supplement, and discussed on pp. 10-11 in the manuscript. While the spatial and temporal variability from the AERONET data are of the same order of magnitude ($\sim 0.02$, depending on the sampling parameters), the spatial variability obtained from the satellite data ($\sim 0.08$) dominates CMU.

4. *The results indicate some difference between the CMU estimated using AATSR and MODIS data. Since AATSR is not as popular a dataset as MODIS. It would be helpful to offer some intercomparison results between these two datasets, e.g., any disagreements in their absolute magnitudes and spatial variability.*

Comparison of AATSR and MODIS AOD products (on a global scale) has been done elsewhere (e.g. de Leeuw et al. 2015), and the results are in agreement in general. In this study, the correlations between collocated AOD values of AATSR and AERONET are similar to those of MODIS and AERONET for most of the sampling parameters (Figs. 4 and 9). The spatial variability of AERONET AOD correlates slightly stronger with the corresponding values from MODIS than from AATSR (Figs. 6 and 9).

We can also compare the AOD values from AATSR and MODIS for the study area and period by regridding the MODIS 10 km product data to the AATSR ADV 0.1 degree grid. We find that the collocated AOD values of AATSR and MODIS agree well with R=0.89 (Fig. 2 below, now added to the Supplement as Fig. S10). The day-to-day changes in AOD variability for the full study area are tracked in a similar fashion by both AATSR and MODIS, as shown in Fig. 3 below. We have replaced Fig. 7 in the manuscript by this figure, so that MODIS is included in the comparison. We have also updated Table S3 in the Supplement to include MODIS data. MODIS overestimates the variability less than AATSR and has slightly better correlation with AERONET. We have added a brief discussion of the inter-satellite comparison to the manuscript (Section 4.3, p. 15). We note that to properly compare the spatial variability of AOD between the two satellite instruments would require more careful sampling, and is beyond the scope of this study. Here we mainly wanted to demonstrate that the methods applied to the AATSR data are in principle applicable to other satellite instruments as well.

Please note that while reproducing Fig. 7, a small error in calculating the temporal collocations was discovered and corrected. This changes the daily collocated AERONET values and the corresponding correlation coefficients slightly, as can be seen in the updated Fig. 7 and Table S3. This error only affected the daily collocation for the whole study area (time series), and has no effect on the other results shown in the manuscript.

[Figure]

Figure 2: Comparison of collocated AATSR ADV and MODIS 10 km AOD values at 555 nm for the DRAGON 2011 campaign.

[Figure]

Figure 3: Day-to-day changes in AOD and spatial AOD variability for the full study area for AATSR, MODIS and AERONET. Note that here we limit the consideration to days when AATSR has data over the area; for MODIS there is data for more days than shown here.

5. *Figures 4, 6, and 7-9 seem a bit difficult to read. I suggest increase the line weights and fond sizes a little bit for clearer presentation.*

   We have increase the font sizes and line widths, and will consult the Editor about options for increasing the image sizes.

| Ind | Site name | URB | $N_m$ | AATSR | | AERO | | R | $\Delta\tau$ |
|---|---|---|---|---|---|---|---|---|---|
| | | | | AOD | $\sigma$ | AOD | $\sigma$ | | |
| 1 | DRAGON_ABERD | 1 | 9 | 0.31 | 0.08 | 0.22 | 0.02 | 0.96 | 0.08 |
| 2 | DRAGON_ANNEA | 1 | 8 | 0.20 | 0.07 | 0.19 | 0.02 | 0.98 | 0.01 |
| 3 | DRAGON_ARNCC | 1 | 10 | 0.15 | 0.05 | 0.15 | 0.01 | 0.98 | 0.00 |
| 4 | DRAGON_ARNLS | 1 | 7 | 0.19 | 0.05 | 0.20 | 0.01 | 0.97 | -0.01 |
| 5 | DRAGON_Aldino | 0 | 13 | 0.27 | 0.07 | 0.18 | 0.02 | 0.77 | 0.09 |
| 6 | DRAGON_BATMR | 1 | 10 | 0.23 | 0.08 | 0.19 | 0.02 | 0.97 | 0.04 |
| 7 | DRAGON_BLDND | 1 | 11 | 0.22 | 0.08 | 0.24 | 0.02 | 0.97 | -0.02 |
| 8 | DRAGON_BLLRT | 1 | 8 | 0.22 | 0.09 | 0.16 | 0.01 | 0.94 | 0.06 |
| 9 | DRAGON_BLTCC | 1 | 8 | 0.20 | 0.07 | 0.22 | 0.01 | 0.98 | -0.02 |
| 10 | DRAGON_BLTNR | 1 | 8 | 0.20 | 0.05 | 0.21 | 0.02 | 1.00 | -0.01 |
| 11 | DRAGON_BOWEM | 1 | 11 | 0.19 | 0.04 | 0.22 | 0.02 | 0.83 | -0.03 |
| 12 | DRAGON_BTMDL | 0 | 9 | 0.22 | 0.09 | 0.22 | 0.03 | 0.94 | 0.00 |
| 13 | DRAGON_Beltsville | 1 | 10 | 0.26 | 0.06 | 0.25 | 0.02 | 0.98 | 0.02 |
| 14 | DRAGON_CLLGP | 1 | 10 | 0.17 | 0.05 | 0.15 | 0.02 | 0.91 | 0.02 |
| 15 | DRAGON_CLRST | 0 | 10 | 0.27 | 0.07 | 0.21 | 0.01 | 0.90 | 0.06 |
| 16 | DRAGON_CPSDN | 0 | 1 | NaN | NaN | 0.53 | 0.00 | NaN | NaN |
| 17 | DRAGON_EDCMS | 0 | 14 | 0.18 | 0.04 | 0.24 | 0.02 | 0.98 | -0.06 |
| 18 | DRAGON_ELLCT | 1 | 3 | 0.17 | 0.06 | 0.16 | 0.01 | NaN | 0.01 |
| 19 | DRAGON_EaglePoint | 0 | 4 | 0.33 | 0.04 | 0.33 | 0.01 | 1.00 | -0.00 |
| 20 | DRAGON_Edgewood | 0 | 14 | 0.29 | 0.08 | 0.25 | 0.03 | 0.86 | 0.05 |
| 21 | DRAGON_Essex | 1 | 13 | 0.22 | 0.08 | 0.21 | 0.02 | 0.96 | 0.01 |
| 22 | DRAGON_FLLST | 1 | 12 | 0.26 | 0.05 | 0.23 | 0.02 | 0.97 | 0.03 |
| 23 | DRAGON_FairHill | 0 | 13 | 0.27 | 0.04 | 0.24 | 0.04 | 0.98 | 0.03 |
| 24 | DRAGON_KentIsland | 0 | 10 | 0.22 | 0.05 | 0.18 | 0.02 | 0.98 | 0.05 |
| 25 | DRAGON_LAUMD | 1 | 5 | 0.21 | 0.03 | 0.18 | 0.02 | 0.98 | 0.03 |
| 26 | DRAGON_MNKTN | 0 | 10 | 0.24 | 0.05 | 0.23 | 0.02 | 0.94 | 0.01 |
| 27 | DRAGON_OLNES | 1 | 12 | 0.24 | 0.05 | 0.20 | 0.01 | 0.92 | 0.04 |
| 28 | DRAGON_ONNGS | 1 | 10 | 0.19 | 0.06 | 0.19 | 0.02 | 0.98 | 0.00 |
| 29 | DRAGON_PATUX | 0 | 9 | 0.16 | 0.04 | 0.18 | 0.02 | 0.95 | -0.01 |
| 30 | DRAGON_Padonia | 1 | 7 | 0.19 | 0.08 | 0.22 | 0.02 | 0.60 | -0.03 |
| 31 | DRAGON_Pasadena | 0 | 8 | 0.22 | 0.08 | 0.18 | 0.03 | 0.96 | 0.03 |
| 32 | DRAGON_PineyOrchard | 0 | 7 | 0.19 | 0.05 | 0.19 | 0.02 | 0.99 | 0.00 |
| 33 | DRAGON_Pylesville | 0 | 9 | 0.26 | 0.04 | 0.23 | 0.02 | 1.00 | 0.03 |
| 34 | DRAGON_SPBRK | 1 | 12 | 0.22 | 0.06 | 0.18 | 0.02 | 0.98 | 0.03 |
| 35 | DRAGON_UMRLB | 1 | 8 | 0.17 | 0.05 | 0.17 | 0.04 | 0.97 | 0.01 |
| 36 | DRAGON_WSTFD | 0 | 8 | 0.28 | 0.07 | 0.20 | 0.01 | 0.86 | 0.08 |
| 37 | DRAGON_Worton | 0 | 8 | 0.31 | 0.06 | 0.24 | 0.02 | 0.96 | 0.07 |
| 38 | GSFC | 1 | 9 | 0.15 | 0.03 | 0.15 | 0.02 | 0.99 | -0.00 |
| 39 | MD_Science_Center | 1 | 3 | 0.17 | 0.05 | 0.17 | 0.00 | NaN | 0.00 |
| 40 | SERC | 0 | 6 | 0.24 | 0.04 | 0.30 | 0.02 | 0.96 | -0.05 |
| 41 | UMBC | 1 | 10 | 0.19 | 0.06 | 0.23 | 0.02 | 0.95 | -0.03 |
| 42 | UMBC_temp | 1 | 10 | 0.19 | 0.06 | 0.19 | 0.02 | 0.94 | 0.00 |
| | Average | 0.6 | 9.0 | 0.22 | 0.06 | 0.21 | 0.02 | 0.94 | 0.03 |
| | Aver. (urban) | 1.0 | 9.0 | 0.20 | 0.06 | 0.20 | 0.02 | 0.94 | 0.02 |
| | Aver. (non-urban) | 0.0 | 9.0 | 0.25 | 0.06 | 0.24 | 0.02 | 0.94 | 0.04 |

Table 1: AOD comparison between AATSR and individual AERONET sites in the study area for the study period. Column 'URB' is 1 for urban sites, 0 for non-urban sites. Column '$N_m$' gives the number of matches i.e. the number of AATSR overpasses in cloud-free conditions. The averages over the columns are calculated for all sites, for the 25 urban sites, and for the 17 non-urban sites, respectively. The sampling parameters used in this comparison are d=0.20 °, $\Delta t$=1.00 h.

---

## Author Comment (AC2) · 11 Jan 2018

We thank the Referee for the positive feedback and constructive comments. The numbered comments are answered below and the changes to the manuscript are indicated in a separate pdf file.

1. *Figures in general have small text, which makes the legends difficult to read at the size they appear in the manuscript. Consider increasing the font sizes.*

   We have increased the font sizes and line widths for the figures with many lines, and will consult the Editor about options for increasing the image sizes.

2. *Page 6, lines 20-33. How does the range of the parameters used for previous*

[Figure]

*work compare to the range used in this study?*

The range of sampling parameters employed in this study covers the range of parameters used in previous studies. This is now added to the manuscript (p. 7, line 1). We find that the previously used parameters are reasonable.

3. *Page 10, lines 6-7. Minimum sample number is another parameter that affects collocations. Do you know whether it's sensitive to the number of satellite pixels needed to form a match?*

The considerable effect of minimum sample number is discussed on p. 13, lines 10-14, and shown in Figs. S3 and S4 in the Supplement. The number of samples is a potential indicator of cloudiness in the area or in the time window close to the overpass and appears to affect the AOD (as discussed in the manuscript, p. 9). A more detailed study on the effect of the cloud proximity on the CMU would be interesting, but that is beyond the scope of this study.

4. *Page 14, line 11. The systematic positive bias in the MODIS data is specific to Terra, and discussed in Levy et al. (2013).*

We agree that the systematic positive bias for MODIS Terra is well known in the community. We have added the reference to Levy et al. (2013) to the sentence.

---

## Author Comment (AC3) · 11 Jan 2018

We thank the Referee for the positive feedback, constructive comments, and for improving the language of the manuscript. The general and specific comments are answered below and the changes to the manuscript are indicated in a separate pdf file.

**General comment**

*The DRAGON data set allows quantification of 'real' spatial and temporal variabilities of AOD, thus quantification of 'real' CMU (as opposed to the simple standard deviation of AOD shown in this paper). I believe that adding discussions on the real CMU and potential limitation of the satellite-derived CMU will improve the quality of the paper.*

[Figure]

We recognize the value of the DRAGON data set as a rare source of information on the 'real' AOD variability. The use of this ground-base data is crucial in evaluation of the satellite based CMU estimate, and we have tried to emphasize this throughout the manuscript. The limitations of the satellite-based CMU estimate with respect to the ground-based data is brought up several times in the text: in the abstract (p. 1, lines 10-18), in the introduction (p. 2, lines 3-10), in the methods section 3.2 (p. 7, lines 20-26), again in the results section 4.2 (p. 10, lines 11-17), and finally in the conclusions (p.16, lines 13-15).

The aim of the paper is to provide a simple estimate of CMU, obtainable globally from the satellite data alone, and hence the simple metric (standard deviation of AOD in a sample) was chosen. We used the same metric (standard deviation of AOD for a sample of sites or a sample of measurements for a given site) for the 'real' CMU obtained from the AERONET sites for consistency.

The standard deviation of AOD from a limited number of observations in a given area may not be the best metric for the CMU. We are aware of the airborne data from DISCOVER-AQ, which shows that the AOD can vary by more than 0.2 within 10 km (Munchak et al. 2013; this is cited on p. 2, lines 31-34 of this manuscript). Thus, even the dense network of AERONET sites provided by the DRAGON campaign may not capture the aerosol variability in full detail. Also, as Anonymous Referee #1 pointed out, the use of a symmetric circular area for sampling may not be the optimal choice due to aerosol transport and varying topography. Surely there is room for more detailed studies of aerosol variability and CMU, possibly using high resolution aerosol dispersion models, but that is beyond the scope of this study.

**Specific comments**

- *Page 1, line 13: the local AOD variability values correlate only weakly for short distance → the spatial variability correlates only weakly with that of AERONET*

*for short distances*

Corrected as suggested.

- *Page 1, line 18: the total uncertainty estimates → the total uncertainty estimates including the CMU*

  Corrected as suggested.

- *Page 2, line 14: based on the them → based on them*

  Corrected as suggested.

- *Page 3, line 10: and conclude that → and concluded that*

  Corrected as suggested.

- *Page 3, line 21 and some other places: on the average → on average*

  Corrected, also on p. 11, line 2.

  On p. 10, line 7 we have replaced 'The AATSR AOD variability ($\sigma_{\mathrm{AATSR}}$) is much larger than the corresponding AERONET value ($\sigma_{\mathrm{AERO}}^{\mathrm{NEAR}}$) on the average' by 'The AATSR AOD variability ($\sigma_{\mathrm{AATSR}}$) is much larger on average than the corresponding AERONET value ($\sigma_{\mathrm{AERO}}^{\mathrm{NEAR}}$)'.

- *Page 7, lines 12-19: It is recommended to move this paragraph to the end of the section and modify to reflect connections between the statistics of this paragraph (number of data points, mean, standard deviation, etc.) and those of the following paragraphs (N-AATSR, N-AERO, sigma-AATSR, sigma-AERO, etc.).*

  We have moved the paragraph to the end of the section and added the terms $\sigma_{\mathrm{AATSR}}$, $\sigma_{\mathrm{AERO}}$, $\sigma_{\mathrm{AERO}}^{\mathrm{NEAR}}$, $N_{\mathrm{AATSR}}$, $N_{\mathrm{AERO}}$, and $N_{\mathrm{NEAR}}$ where appropriate.

- *Page 9, line 4 and other places: effect . . . to → effect . . . on*

  Corrected, also on p. 7, line 4 and in Fig. 8 caption.

- *Page 16, line 9: sampling distance increases → sampling distance increases further*

  Corrected as suggested.

---

## Author Comment (AC5) · 12 Jan 2018

**Response to Short Comment**

**Collocation Mismatch Uncertainties in Satellite Aerosol Retrieval Validation**

Timo H. Virtanen, Pekka Kolmonen, Larisa Sogacheva, Edith Rodríguez, Giulia Saponaro, and Gerrit de Leeuw
Finnish Meteorological Institute, Helsinki, Finland

We thank Dr. Schutgens for the positive feedback and the interesting questions.

- *The authors conclude that there is an optimal sampling distance of 0.3-0.4 degrees for such validations. That should give some confidence in 1 by 1 degree L3 products.*

  Indeed, the AOD comparison results indicate that a larger sampling area, approximately the size of a L3 pixel, gives better agreement between the satellite and AERONET than a single-pixel comparison. Also, the AOD variability is better captured at the $\sim 1°$ scale, which gives some credibility to L3 uncertainty estimates.

- *I wonder though why correlations drop off much faster for MODIS data (Fig 9) than for AATSR data (Fig 4)? According to the authors, this drop-off is due to the natural variability in an AOD field (i.e. not related to retrieval errors). Wouldn't you expect MODIS and AATSR data to show similar results?*

  The main problem here may be the different scaling of the y-axis in Figs. 4 and 9. In Fig. 1 below we have rescaled Fig. 7 a) from the manuscript so that the y-axis has the same scale as in Fig. 9 a). Note that the y-axis position is still different for AATSR and MODIS, but in both panels below the y-axis covers a correlation coefficient range of 0.35 (from 0.92 to 0.955 for AATSR and from 0.935 to 0.97 for MODIS). On this scale, the drop in AATSR correlation with increasing sampling distance is better visible, although it is slower than for MODIS. However, the correlation coefficients remain slightly higher for MODIS than for AATSR. The main difference between the instruments seems to be that for the smallest sampling distances AATSR does not agree so well with AERONET.

  We will now point out the difference in the y-axis scales in the paper.

[Figure]

Figure 1: Comparison of AATSR (a) and MODIS (b) correlation coefficients with AERONET for various sampling parameters. Here the y-axes have the same scale in both panels.

- *We recently studied the representativity of observations using high resolution models: https://www.atmos-chem-phys.net/17/9761/2017/ . One interesting aspect is that modelled fields show high correlation among neighbouring points while your satellite-AERONET comparison shows very low correlation amongst neighbouring points (i.e. short sampling distance). As the authors themselves conclude this points to large random errors in the satellite product that average out when using larger sampling distances.*

  The high noise in AATSR data which averages out for larger sampling distances is certainly a point we want to address in future studies, possibly utilizing the high resolution (1 km) AATSR retrievals. The issue may be related to cloud screening: a fragment of cloud missed by the cloud screening algorithm will cause an erroneous high AOD value in the satellite data. As discussed in the manuscript, the number of valid AATSR pixels ($N_{\mathrm{ADV}}$) in the sampling area is an indirect measure of the cloudiness in the area. Fig. 2 shows that by requiring at least three AATSR pixels in the sampling area improves the comparison considerably for the small sampling distances.

[Figure]

Figure 2: Comparison of AATSR (a) and MODIS (b) correlation coefficients with AERONET. Here we have required at least 3 satellite pixels in the sampling area and at least 3 AERONET observations in the sampling time window.